# Correlation of Faecal Egg Counts with Clinical Parameters and Agreement between Different Raters Assessing FAMACHA©, BCS and Dag Score in Austrian Dairy Sheep

**DOI:** 10.3390/ani13203206

**Published:** 2023-10-13

**Authors:** Floriana Sajovitz, Isabella Adduci, Shi Yan, Sandra Wiedermann, Alexander Tichy, Anja Joachim, Thomas Wittek, Barbara Hinney, Katharina Lichtmannsperger

**Affiliations:** 1Clinical Unit of Ruminant Medicine, University Clinic for Ruminants, Department for Farm Animals and Veterinary Public Health, University of Veterinary Medicine Vienna, Veterinärplatz 1, 1210 Vienna, Austria; 2Institute of Parasitology, Department of Pathobiology, University of Veterinary Medicine Vienna, Veterinärplatz 1, 1210 Vienna, Austria; 3Bioinformatics and Biostatistics Platform, Department of Biomedical Sciences, University of Veterinary Medicine Vienna, Veterinärplatz 1, 1210 Vienna, Austria

**Keywords:** Austria, faecal egg count, inter-rater agreement, body condition score, FAMACHA©, dag score, trichostrongylids, gastrointestinal nematodes

## Abstract

**Simple Summary:**

The objectives of this study were to evaluate the associations between the faecal egg count for strongylids and the clinical parameters in lactating dairy ewes. In addition, the agreement of three raters regarding the clinical examination was assessed. For this purpose, a total of 1195 dairy ewes from 16 farms were included. FAMACHA©, BCS and dag scores were determined and a Mini-FLOTAC was performed on individual samples, which were then pooled for larval culture at the farm level according to the number of lactations. Trichostrongylids were found in 95% of the investigated samples. The BCS was negatively correlated with the faecal egg count, while the FAMACHA© score showed a slightly positive correlation. The dag score did not show significant associations. A minority (25%) of the flock shed the majority of eggs (47% to 84%). It was concluded that clinical scoring does not allow us to extrapolate to faecal egg excretion. The agreement was moderate to good between different raters, showing that training on clinical parameters is recommended for assessment in sheep. From a clinical perspective, both faecal egg counts and clinical parameters should be used simultaneously as separate tools to detect major egg shedders (contaminating the pasture) and sheep suffering from strongylidosis.

**Abstract:**

Gastrointestinal nematodes, most notably trichostrongylids, are known to cause significant losses in sheep production. Previous studies have shown that monitoring parameters (e.g., FAMACHA©, BCS, dag score) change with increasing egg excretion. These parameters are well known and frequently used for targeted selective treatment. Based on the willingness to participate in this study (based on a previous questionnaire distribution among sheep farmers in Austria) we investigated the associations between faecal egg counts and the FAMACHA©, BCS, and dag scores of 1195 dairy ewes. Faecal samples were analysed using the Mini-FLOTAC technique I and larval culture. Three raters assessed the FAMACHA©, BCS, and dag scores in sheep to calculate the inter-rater agreement and intraclass correlation coefficient. The responses to the questionnaire of 23 farms were used for the evaluation, of which 16 farms were visited. Trichostrongylid eggs were detected in 95% of the faecal samples. The BCS was negatively correlated with the eggs per gram of faeces (EpG) (r = −0.156; *p* < 0.001) and the FAMACHA© score was slightly positively correlated with EpG (r = 0.196; *p* < 0.001). A small proportion of sheep (25%) shed the majority of eggs (47% to 84%). A moderate to good agreement for the parameters was found between the raters. In conclusion, the clinical parameters showed only weak correlations with faecal egg counts, and we confirmed that a minority of the flock is responsible for the majority of the pasture contamination with trichostrongylid eggs. Clinical raters should be trained before a study to increase the agreement between them.

## 1. Introduction

Trichostrongylid infections are considered to be a major health threat in small ruminants worldwide. Significant economic losses are caused due to reduced feed intake, lower growth rates, and poor performance [1,2,3,4]. In Austria, the annual costs due to infections with helminths are estimated at EUR 936,257 and EUR 1,225,383 for dairy and meat sheep, respectively [4].

Control of trichostrongylid infections with routine deworming procedures in the absence of an efficacy evaluation, but also due to overuse or misuse of antiparasitic drugs, have led to increasing anthelmintic drug resistance, resulting in a failure in treatment. Control strategies that strongly select for resistant worms are frequently applied [5,6]. The fast and widespread development of anthelmintic resistances is an emerging challenge for the livestock industry [7,8]. Recent investigations conducted in Austrian sheep and goat flocks have shown a reduced efficacy in various anthelmintic drugs [9,10,11]. Therefore, the implementation and application of alternative methods and individual farm monitoring concepts are crucial to slow the development of efficacy loss for anthelmintic drugs. Integrated parasite management is a sustainable approach in which treatment and management are combined [12,13]. Targeted selective treatment (TST) is based on the evaluation of clinical parameters (e.g., FAMACHA©, Body Condition Score, dag score) to identify and treat sheep that would benefit the most from treatment. Additionally, sheep can also be selected for TST based on faecal egg counts to target animals that are responsible for most of the environment contamination [14,15]. However, individual faecal sampling of sheep flocks is not feasible in most cases, especially on a large scale, as this is cost-intensive and time-consuming. In terms of TST, only a part of the flock is treated and susceptible worms remain in untreated animals (refugium) to avoid selection for anthelmintic resistance [12,14,16]. Clinical assessment, but also the FEC and faecal egg count reduction test (FECRT), should be an integral part of the farm routine, as they can lead to an improvement in animal health. These assessments are easy to implement and in most cases, (especially clinical examinations) are cost-effective [12,15,17,18,19]. Clinical parameters are postulated to be associated to the number of excreted eggs [20,21,22]. However, this may depend on the prevailing nematode species, the host, age, nutrition, and environmental factors [19,23]. *Haemonchus contortus*, predominantly parasitizing small ruminants, is known to be blood-sucking, causing paleness of the mucosal membranes, while the faeces tend to retain a physiological consistency [24]. Therefore, monitoring flocks with the FAMACHA© scoring system that evaluates the colour of mucous membranes is a practical way in the field to diagnose anaemia due to infections with this roundworm species [25]. The tool is frequently used in decision-making for TST [26].

However, as gastrointestinal nematode infections are typically caused by mixed species, including trichostrongylids parasitizing the small intestines, diarrhoea should be taken into account [12,27,28]. For this, the dag score offers a straightforward classification of faecal soiling [28]. As food intake and digestion may be impaired by infections with gastrointestinal nematodes (GIN), affected individuals may suffer from severe weight loss and poor body condition [1,2,20,29]. This can be monitored using the body condition scoring (BCS) system [30]. By determining the faecal egg count, it is possible to identify high egg shedders within the flock that contaminate the environment with parasite stages, increasing the infection pressure for the flock. In addition, possible anthelmintic resistance (AR) can be detected by faecal egg counts, which should decrease by at least 95% after deworming [31,32].

The western part of Austria is a mountainous area where there are primarily small-scale family-owned farms. Austria has a sheep population of 402,345 sheep of which ~30% are kept on organic farms (calendar year: 2022) [33]. In 2022, meteorological data from Austria showed that the average ambient temperature increased on average by +2.3 °C in comparison to the years 1961–1990 [34]. In general, the average ambient temperature was 8.1 °C within Austria. The warmest city in the flatland region showed an average temperature of 13 °C. The annual precipitation reached from 400 mm in one defined region in Lower Austria to 2400 mm in the Alps. The average annual precipitation was 940 mm [35].

The objectives of the present study were to determine the faecal egg count (FEC), FAMACHA©, BCS, and dag score in lactating sheep and to calculate the associations between the clinical parameters and the FECs, as well as to assess the agreement of the clinical parameters of three different raters. We hypothesized that faecal egg counts are significantly correlated with BCS, FAMACHA©, and dag score, and the agreement between the three raters is high.

## 2. Materials and Methods

### 2.1. Ethical Consideration

This study was approved by the Ethics and Welfare Committee of the University of Veterinary Medicine, Vienna in accordance with the University’s guidelines for Good Scientific Practice and the national animal protection legislation. As only non-invasive procedures were performed in the study, governmental approval was not required.

### 2.2. Questionnaire and Farm Selection

In December 2021, a questionnaire was distributed to all registered Austrian sheep breeders via the Austrian Sheep and Goats Breeding Association (ÖBSZ) in order to collect information about the farms, animals, and the farm management for a pre-assessment and history of the farms. The questionnaire included a part where farmers were invited to participate in the clinical part of this study. In addition to the consent to participate via the questionnaire, dairy sheep farmers that contacted the University Clinic for Ruminant directly for a farm visit were also approached to participate in the questionnaire and the clinical study. The online survey was created using SurveyMonkey.de and contained 76 questions. Different types of questions were included, e.g., yes/no, single- and multiple-choice questions, open-ended questions, and questions with skip logic with a progression through the survey varying depending on the answer. The questionnaire consisted of nine subsections with general and specific questions about: (A) farm (1–13), (B) animals and housing (14–22), (C) biosecurity, hygiene, and animal purchases (23–32), (D) feeding (33–36), (E) pasture management (37–51), (F) parasite management (52–65), (G) diagnostics and efficacy checks (66–72), (H) estimation (from the farmer’s point of view) of parasitic diseases on the farm (73–75), and (I) voluntary participation in the clinical study (76).

### 2.3. Farms and Animals

The farm visits were performed between February and June 2022. The included farms (n = 16) were located in Upper Austria (n = 5), Lower Austria (n = 3), Carinthia (n = 4), Styria (n = 2), Salzburg (n = 1), and Tyrol (n = 1). The number of included sheep per farm ranged from 17 to 161 (median = 81). The only selection criterion was that lactating sheep had to be in the first two-thirds of lactation at the time of sampling to have a homogenous set of samples. In total, 1299 ewes of various ages and three different breeds (Lacaune, Carniolan Stone Sheep, and East Friesian sheep) and mixes of these were initially included. During the farm visits, all lactating sheep were sampled by rectal faecal sampling. Individual faecal samples were collected in 100 ml plastic collection cups marked with the ear tag identification number of the sheep. Samples were transported on ice in a polystyrene box to the Institute of Parasitology, Vetmeduni Vienna, Austria for further examination. If an animal had no faeces in the *ampulla recti* or the animal was obviously ill due to other reasons (e.g., mastitis or severe lameness) the animal was excluded (n = 104) from the study. Hence, individual faecal samples from 1195 sheep were further analysed.

### 2.4. On-Farm Assessment of Clinical Parameters

The assessments of the clinical parameters (FAMACHA©; BCS; dag score; n = 1195) were carried out by the first author and/or by one of the two final year veterinary students who were instructed prior to the visits. A subset of sheep was examined by all three investigators. The sheep were examined by the raters at the same time.

### 2.5. On-Farm FAMACHA© Score Assessment

All sheep were individually assessed using the FAMACHA© scoring system [25] by the first author (FS) and/or a final year veterinary student. It is a five-point-scale scoring system that corresponds to the colour of the ocular mucous membranes for the detection of anaemic animals [25]. Prior to the investigation, an information sheet about FAMACHA© scoring [36] was provided to all observers. The assessment was performed by the first author (FS) or one of the final year veterinary students under good light conditions. In addition, both eyes were examined with the higher score being recorded.

### 2.6. On-Farm BCS Assessment

The five-point body condition score assessment [30,37] method especially designed for sheep was implemented in a modified version in the present study. In brief, sheep were scored with intermediated steps, therefore nine scores were possible (score 1 = emaciated; score 1.5 = between emaciated and thin; score 2 = thin; score 2.5 = between thin and average; score 3 = average; score 3.5 = between average and fat; score 4 = fat; score 4.5 = between fat and adipose; score 5 = adipose). Drawings of the different BCS were used to standardize the assessment.

### 2.7. On-Farm Dag Score Assessment

A range from 0 and 5 was implemented for the dag score. Score 0 represented no evidence of faecal soiling and a score of 5 represented a highly soiled animal with dag extending below the hocks [28].

### 2.8. Examination of Faecal Samples

#### 2.8.1. Mini-FLOTAC

Individual faecal samples from dairy sheep were analysed quantitatively by Mini-FLOTAC [38]. The lower limit of detection of this method is five eggs per gram (EpG) of faeces [39]. All samples were examined on the same or the following day of the farm visit and were stored before examination at 4°C. Samples were weighed (5 g faeces) and mixed with 45 ml of a saturated saline solution (density 1.18 g/mL) using pistil and mortar to a homogeneous mixture and sieved (Ø 1.3 mm) into a plastic cylinder. By using a magnetic stirrer (IKA-COMBIMAG REO, Janke & Kunkel GmbH u. Co. KG, Staufen, Germany), an equal distribution of the eggs was ensured during removal of the aliquots for counting. Samples were aspirated with a disposable 2 mL pipette and transferred to the Mini-FLOTAC system. After 10 min of resting on a flat surface, the disks were rotated with the key. If air bubbles were visible in the analysis chambers the sample was discarded and the steps were repeated. Faecal samples were quantitatively evaluated under the light microscope (Eclipse Ci-S, 100×magnification, Nikon, Vienna, Austria). The EpG was calculated with the following formula: $EpG=egg\;counts\;in\;both\;chambers\times \frac{45+5}{5\times 2}$.

#### 2.8.2. Larval Cultures

For larval cultures, samples were pooled according to the number of lactations. Samples that contained no eggs in the Mini-FLOTAC counting were excluded. For a single pool, at least five samples were used. If this was not possible as no faeces were left or it was a FEC of less than five, samples from animals from the same farm in the same lactation period were pooled for larval culture. If the lactation number of the sheep was unknown, the faecal samples were summarized in one pooled sample (farm-level).

Faecal samples were homogenised with a mixer, mixed with vermiculite, and moistened with water. The mixture was incubated at 26 °C for at least two weeks and stirred daily with a spatula to ensure ventilation. At the end of incubation, the cultures were topped up with water, covered with a Petri dish, and everted. After approximately 24 h, the third-stage larvae (L3) were collected with the liquid in the dish and stored in 50 mL tubes at 4 °C.

#### 2.8.3. Larval Differentiation

For differentiation, the larvae were transferred to a glass slide using a pipette, covered with a cover slip, and classified under a light microscope (Eclipse Ci-S, Nikon, Tokyo, Japan) with 100–400× magnification. Morphological measurements were carried out using a GRYPHAX^®^ SUBRA camera attached to the microscope and the GRYPHAX^®^ software program version 2.1.0.724 (Jenoptik, Jena, Germany). For genus classification, published keys were used [40,41].

#### 2.8.4. Sedimentation (Benedek)

Samples were pooled (farm-level) for the detection of liver fluke (*Dicrocoelium dentriticum, Fasciola hepatica*). The faeces (5–10 g) was mixed with 20 mL of tap water into a homogeneous suspension using a pistil and mortar, sieved (Ø 1.3 mm), and placed in a 250 mL beaker, which was then filled to the brim with tap water. When the sediment was visible at the bottom of the beaker, the supernatant was decanted. This was repeated at least three times until the supernatant was transparent. The sediment was then placed in a Petri dish to be examined under a stereo microscope (SZH10 Research stereo, Olympus, Hamburg, Germany) with 100–400× magnification.

#### 2.8.5. Baermann–Wetzel Technique

For the Baermann–Wetzel technique, the same pooled samples were used as for the detection of liver fluke. About 20 g of faeces were wrapped in two layers of gauze and placed in a sieve (Ø 1.3 mm) in a funnel with a clamped tube at the conical end. The funnel was filled with tap water to just below the rim. To give the larvae time to migrate, the funnel was left overnight (for at least 6 h). The sediment was collected in a Petri dish by opening the clamp and was then examined under a stereo microscope (SZH10 Research stereo, Olympus, Hamburg, Germany) with 100–400× magnification.

### 2.9. Statistical Analysis

Data were collected and summarized in Microsoft Excel^®^ 2016. The complete dataset was transferred to IBM SPSS^®^ Statistics Version 28 (IBM, New York, NY, USA) for further statistical analysis. The results of the three raters (rater 1vet = veterinarian = reference; rater 2stud = final year veterinary student 1; rater 3stud = final year veterinary student 2) were encoded for the three clinical parameters. The FAMACHA© score had five categories (1; 2; 3; 4; 5), the BCS score had nine categories (1; 1.5; 2.0; 2.5; 3.0; 3.5; 4; 4.5, 5.0), and the dag score had six categories (0; 1; 2; 3; 4; 5). The EpG values were logarithmically transformed using the log function in SPSS (log10). They were tested for normal distribution using the Kolmogorov–Smirnov test with the post-hoc Lilliefors correction. The EpG values were not normally distributed. Descriptive statistics were carried out for the EpG values including the 10-25-50-75-90 percentiles. Additionally, the cumulative egg shedding was calculated per farm (summary of EpG by the sheep per farm). The infection status of the sheep was categorized as below the limit of detection (<5 EpG), 5 to 500 EpG (low), 501 to 1000 EpG (moderate), or >1000 EpG (high) excretion, as described by [42]. Additionally, the EpG values were split in high (≥797 EpG) and low (<797 EpG) shedders using the mean EpG value of all 1195 sheep of 797 EpG. For the FAMACHA© score, the BCS, and the dag score the median, maximum, and minimum values were included. The agreements between the three raters were expressed using weighed kappa. Kappa values can range from 0 to 1 and were interpreted as follows: ≥0.81 very good agreement; 0.61 to 0.80 good agreement; 0.41 to 0.60 moderate agreement; 0.21 to 0.40 fair agreement; and ≤0.2 poor agreement [43]. The overall agreement of all raters for each clinical parameter was expressed as an intra-class correlation coefficient (ICC), for which values >0.8 were interpreted as good agreement. To assess the correlation between the EpG values (continuous variable) and the clinical parameters FAMACHA©, BCS and dag score (nominal/ordinal variables), the bivariate Spearman Rank correlation coefficients (correlation between −1 and +1), and its 95% confidence intervals (95%CI) were calculated. The results from all clinical assessment from all 1195 sheep (rater 1 to rater 3) were included. *p*-values < 0.05 were considered as statistically significant. Farms highly positive for *H. contortus* (i.e., >50% *H. contortus* in the larval differentiation according to the differentiation of larvae described by Knoll and co-workers, 2021 [40]) were further investigated to calculate the Spearman Rank correlation between the FAMACHA© score, the BCS, and the dag score with the EpG values.

## 3. Results

### 3.1. Participating Farmers and Their Management Practices

In total, 27 farmers responded to the questionnaire, of which 23 completed it. Of these, 18 consented to participate in the study. The total number of respondents varies in some cases per question, since due to the possibility of a free text answer, some answers could not be assigned and were excluded from the evaluation. Of the respondents (n = 23), 18 produced according to the Austrian organic farming regulations and 5 as conventional farms. On the majority of the farms, sheep had access to a pasture in summer (91.3%). On 13 farms, the ewes were kept indoors with controlled grazing times depending on the season, while the rest reported year-round grazing (one farm), year-round indoors (two farms), or indoors with turnout and access to a pasture (seven farms). Farmers were asked how they in general proceed when parasite infection was suspected in an individual animal. Of all participating farms (n = 23), ten had an individual faecal sample examined and, in case of a positive result, nine farmers dewormed the affected sheep only, and one dewormed the entire flock/group. Pooled faecal sampling and a subsequent deworming of the entire flock/group if findings are positive was carried out by five farmers. Eight farmers did not obtain faecal sample diagnostics and directly dewormed the affected animal (n = 5) or the entire flock/group (n = 3). Of the latter three, one farmer stated that the entire pasture-flock/group is dewormed when worms are seen macroscopically in the faeces after the grazing season. Thirteen farmers obtained routine faecal sample diagnostics (irrespective of suspected infections), whereby one of them did so for young lambs only. The remaining did not routinely obtain coproscopical results. Only three farmers cleaned the barn after deworming. Regarding the anthelmintic dosage (information provided by 22 farmers), the questionnaire revealed that 12 farmers dewormed according to a visual appraisal without taking the weight of any of the sheep within the flock, while for 7 the heaviest sheep on the farm is used as a guideline. Only three farmers dewormed according to the actual individual animal weight. One farmer did not answer this question. Although the concept of TST was known to 17 farmers, it was applied by 10 farmers only. A multiple-choice question was asked regarding which endoparasites had been diagnosed more frequently in recent years. More than half of the 23 farmers reported trichostrongylids and tapeworm infections (Table 1). *Haemonchus contortus* (information provided by 22 farmers) had already been diagnosed on eight farms and suspected but not confirmed on one farm. The parasite was unknown to five of the twenty-two farmers. The necessity of a follow-up antiparasitic treatment in the past was reported by nine out of twenty-two farms where this information was provided. Farmers were asked how many losses due to parasitic disease they were aware of for the previous year. Here, 22 out of 23 farmers estimated losses of less than 5%, while 1 farmer indicated 5–10%.

### 3.2. FAMACHA© Scores

In total, 1872 FAMACHA© assessments were carried out on 1195 sheep. FAMACHA© scoring was carried out by rater 1vet, rater 2stud, and rater 3stud on 337, 961, and 574 sheep, respectively. The majority of sheep showed a FAMACHA© score of 2 (with score 1 = no anaemia, 5 = severe anaemia) in the clinical assessment, whereof there were clear farm-level differences (see Table 2).

### 3.3. Body Condition Scoring

In total, 1684 BCS assessments were carried out on 1195 sheep. The scoring was carried out by rater 1vet, rater 2stud, and rater 3stud on 1153, 256, and 275 sheep, respectively. The majority of sheep showed BCS levels between 1.5 and 3.0 (Table 2).

### 3.4. Fleece Soiling

In total, 1589 dag score assessments were carried out on 1195 sheep. Dag scoring was carried out by rater 1vet, rater 2stud, and rater 3stud on 1195, 209, and 185 sheep, respectively. The majority of sheep showed a dag score between 1 and 3 (see Table 2).

### 3.5. Quantitative Egg Excretion

In total, 1195 individual faecal samples were examined. The EpG values ranged from 0 to 20,175 EpG (median = 245.0 EpG). Of the examined faecal samples, 5.3% were below the detection limit of 5 EpG. Between farm and within farm variations of faecal egg shedding are shown in detail in Figure 1. Detailed information on egg shedding (descriptive statistics, cumulative egg shedding) per farm is provided in Table 3 and Appendix A. A minority (25%) of the flock shed the majority of eggs (47.3% to 83.8%).

### 3.6. Haemonchus Contortus

The overall prevalence of *Haemonchus contortus* was 59.4% with rates per farm of 24.9–73.9% (Table 4).

### 3.7. Sedimentation

*Dicrocoelium dentriticum* was detected in seven of sixteen farms (44%), *Fasciola hepatica* in two farms (13%). Detailed information is provided in Table 5.

### 3.8. Baermann–Wetzel Technique

Protostrongylidae were found in 13 farms (81%), while no large lung worms were detected. Detailed information is provided in Table 5.

### 3.9. Intra-Class Correlation Coefficient and Inter-Rater Agreement 

The intra-correlation coefficients were 0.678 for FAMACHA© (good agreement), 0.642 for the BCS (good agreement), and 0.669 for the dag score (good agreement). Detailed information on the inter-rater agreement is provided in Table 6.

### 3.10. Correlation between Clinical Parameters and EpG Values

The excretion of strongylid eggs was slightly negatively correlated with the BCS and slightly positively correlated with the FAMACHA© score. There was no significant correlation between the FEC and dag score (Table 7). For sheep herds where *H. contortus* was the predominant trichostrongylid species, the correlation between the FEC and the FAMACHA© score was higher but still weak with 0.233 (lower 95%CI = 0.171, upper 95%CI = 0.294) versus the herds where *H. contortus* was not the predominate species. If extracting sheep with a FAMACHA© score of 3, 4, or 5 (n = 321), the correlation between the FAMACHA© score and EpG was 0.148 (lower 95%CI = 0.036, upper 95%CI = 0.256). Extracting the 1088 sheep with a low BCS (≤3.0), the correlation between the BCS and the EpG was – 0.094 (lower 95%CI = −0.154, upper 95%CI = −0.033; *p* = 0.002). In total, 308 sheep showed a BCS of ≤3 and a FAMACHA© score of 3, 4, or 5. Using these 306 sheep, the correlation between the FAMACHA© score and the EpG was 0.138 (lower 95%CI = 0.023; upper 95%CI = 0.250) and between the BCS and the EpG the correlation was −0.129 (lower 95%CI = −0.241; upper 95%CI = −0.014).

## 4. Discussion

In Austria, sheep farms are primarily small-scale family-owned businesses, with 16,181 farms and 400,664 sheep (Ø 25 sheep per farm) in total [44]. Therefore, it is in most cases feasible to overlook the entire flock and practice health monitoring at an individual level. In previous studies carried out in Austria, the prevalence of gastrointestinal strongylids ranged from 62% and 100% [11,45,46]. For *H. contortus,* prevalences ranged between 19 and 99% in different studies and almost doubled during the pasture season [10,46]. In the present study, trichostrongylids (including *H. contortus*) were detected in all 16 dairy sheep farms (100%) and in 95% (1132) of the included sheep. The on-farm occurrences of *H. contortus* ranged between 25% and 74%. Of all observed L3 larvae, 60% were identified as *H. contortus* [40]. Nevertheless, the results highlight the importance of carrying out regular and comprehensive studies to obtain an update on the situation in specific regions, especially in areas with a high density of small ruminants. In addition, climate may influence the development of *H. contortus* [47,48]. In Austria, the mean temperature in 2022 ranged from 0.6 °C in February to 17.1 °C in June. Compared to the period 1961–1990, the temperature thus deviated by +2.3°C on average. The total precipitation in 2022 was about 940 mm in total with values ranging from 56 mm in February to 150 mm in June (mean 78.6 mm) [34]. In the present study, only a minority of sheep shed the majority of the eggs and therefore contribute the most to environmental contamination. To monitor pasture contamination, FECs can be used [49]. Egg excretion and the worm burden do not correlate well since the host, the environment, and the worm species exert important effects on the number of excreted eggs per female worm [42]. While *H. contortus* is known to be highly fecund with an egg laying capacity of thousands of eggs per day, *Trichostrongylus* spp. produce only a few hundred eggs per day [50]. However, FECs can be very useful to target high egg shedders and subsequently treat these animals which are responsible for most of the pasture contamination with parasite eggs [51].

The second hypothesis of the present study was that there is correlation between the clinical parameters FAMACHA©, BCS, and dag scores in lactating Austrian dairy sheep. According to the manufacturer’s specifications of the FAMACHA© scoring card, sheep with light red to white mucous membranes should be considered to be treated (score 3) or be treated (score 4–5) [25,26]. However, results from a study in Italy showed a low sensitivity in detecting sheep with anaemia using the FAMACHA© system, but correlations with FEC were positive (r = 0.146) [21]. In the present study, there was also only a weak correlation of 0.196 between the FEC and the FAMACHA© score. Additionally, only weak correlations were found for *H. contortus*-positive sheep for the FAMACHA© score (r = 0.233) and the BCS (r = −0.149). However, it must be considered that other parasites, such as the liver fluke, and non-parasitic causes such as ocular irritation or inflammation, poor ventilation, or weather conditions, as well as specific systemic diseases can change the colour of the mucous membranes and may thus influence the visual assessment [25,52]. In the present study, the effect of the liver fluke was not included since there was quantitative data for the individual sheep available. Nevertheless, in two farms the large liver fluke was present and might therefore also affect the clinical appearance of the sheep. We assessed the mucous membranes of both eyes under good light condition in the barn/shed. The results indicate that the FAMACHA© scoring system may not be appropriate for sheep kept (temporarily) indoors, as the air, pollution, or hygiene levels differ compared to outdoors. The FAMACHA© score is not intended to detect high egg shedders, but rather to detect animals at risk of anaemia due to parasitic infection with blood-sucking endoparasites [25]. The sheep were sampled only once in a cross-sectional study design. It might be necessary to perform a longitudinal study and sample repeatedly over a defined time period to account for variability in egg shedding. It is important to underline that the results might be significantly different if another time point, for instance during or after the grazing season, has been chosen for faecal sample collection. It can be hypothesized that the correlation between the FAMACHA© score and the FEC might be higher.

In the present study, different larval differentiation methods returned substantial differences in the percentage of *Haemonchus*-positive sheep and some larvae could not be assigned to a genus due to differences in their length [40,41]. Therefore, the comparison of studies in that respect has to be assessed with care. Molecular methods, such as a sensitive *H. contortus* PCR, could substantially improve the detection of *H. contortus* from faecal samples [53]. However, it is a cost-intensive method that may not enter small ruminant veterinary practice in Austria with small-scale farms.

No correlation was found between the dag score and FEC. High egg counts do not necessarily lead to significant diarrhoea [54]. Other investigations found an association with a high FEC in lambs, but in older sheep the correlation between FEC and diarrhoea changed [22,54]. In the present study, only adult sheep were examined. 

The BCS has been described to be used as a parameter in target-selective treatment [23,55]. The data analysis showed that BCS was slightly negatively correlated with FEC (−0.156; n = 1195 samples; *p* = <0.001). Although results were significant, the correlation was weak. This is in accordance with a previous study that described a correlation between the FEC (with *Teladorsagia circumcincta* and *Trichostrongylus* spp. as the most abundant gastrointestinal strongylids) of lactating ewes and a body condition score of r = −0.163 [20]. Recently a meta-analysis described the effects of gastrointestinal nematode infection on body weight in sheep [3]. Infected animals showed only 85% of the performance (weight gain) compared to uninfected individuals [3]. From the results, however, it could not be concluded that sheep with a poor body condition had a higher egg excretion. From a clinical point of view, emaciated animals have to undergo a proper physical examination with additional parasitological examination to determine the underlying causes [14,25]. Regarding the correlation between FEC and BCS, it has to be stressed that the time of sampling is a significant variable. Therefore, the correlation might be different if the samples are taken during another time period.

We hypothesised that the agreement between different raters assessing FAMACHA©, BCS, and dag scores were high. The results show that the agreement was moderate to good and suggest that from a clinical perspective, intensive theoretical preparation and training are required to obtain reliable results from different examiners/observers in a study. If animal owners/farmers want to implement scoring of clinical parameters as the basis for a decision on anthelmintic treatment or other veterinary interventions, it is highly encouraged to provide training and to regularly re-evaluate the results to achieve reliable results that can be supportive for intervention decisions. In addition, different sheep breeds may have influenced the assessment of BCS. A study carried out in goats described that the FAMACHA© score in combination with the BCS can be used to detect animals at risk for high gastrointestinal nematode infections [56]. Different aspects, such as the development of immunity in adult sheep, the actual worm burden, and worm fecundity must be taken into account when interpreting results [42]. The necessity of the physical examination and subsequent specific diagnostics are highlighted by the results of the present study. A major limitation of this study was that farm visits were only performed once and some sheep were already grazing on pasture for a couple of hours per day while other farms still kept their sheep indoors during the winter break. They mostly had a low body condition score despite previously being dewormed regularly, indicating that other factors contributed to the BCS values. From the presented data it is not feasible to provide any recommendations on a target-selective treatment approach for sheep farms in Austria. Additional faecal sampling points, for instance in autumn following the grazing season, might be necessary to provide a valid recommendation.

## 5. Conclusions

A low BCS had a weak correlation with increased FEC and a higher FAMACHA© score showed a weak correlation with increased FEC; however, in flocks of adult sheep in Austria, high egg shedders cannot be detected by using only these clinical parameters. Other factors that may alter the clinical parameters and/or egg excretion must be considered. In general, sheep shed high amounts of eggs without altered clinical parameters and treatment based only on clinical signs may not include high egg shedders, which are primarily responsible for pasture contamination and reinfections. To reduce pasture contamination, treatment based on FEC is recommended. In addition, reliable clinical scoring appears to be subject to high interrater-variations and requires training for inexperienced examiners.

Finally, it would be beneficial to repeat the study during or after the grazing season to compare the effect of grazing on parameters with the results of the present study, conducted during the winter/spring period.

## Figures and Tables

**Figure 1 animals-13-03206-f001:**
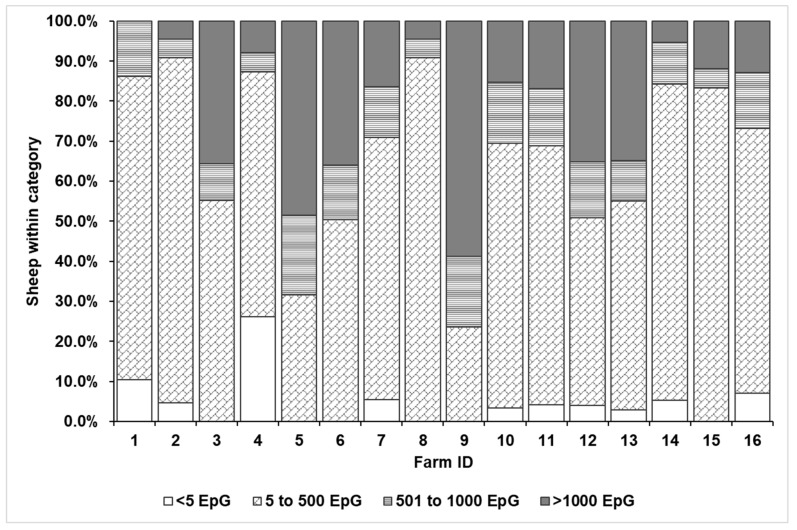
Overview on the infection status of 1195 investigated dairy sheep originating from 16 dairy sheep farms in Austria. The sheep were categorized according to four shedding categories: <5 EpG (below detection limit), 5 to 500 EpG, 501 to 1000 EpG, and >1000 EpG.

**Table 1 animals-13-03206-t001:** Response to the online questionnaire preceding the clinical examinations on farm: multiple-choice question “Which endoparasites have been diagnosed more frequently on your farm in recent years”, n = 23.

Answer Choices	Responses (%)
Trichostrongylids	13 (56.5%)
Tapeworm	13 (56.5%)
Liver fluke	9 (39.1%)
*Coccidia*	9 (39.1%)
Lungworms	4 (17.4%)
Negative for endoparasites	0 (0.0%)
Unknown, no faecal sample examinations	5 (22.7%)

**Table 2 animals-13-03206-t002:** Results of the FAMACHA© anaemia guide (scores 1 = no anaemia to 5 = severe anaemia), BCS (scores 1 = emaciated to 5 = adipose), and dag scoring (scores 0 = no evidence of faecal soiling to 5 = heavily soiled) gathered during the farm visits (n = 16).

Figure	N Sheep	FAMACHA©			BCS			Dag Score		
		Quartiles			Quartiles			Quartiles		
		25	50	75	25	50	75	25	50	75
1	29	2.0	2.0	3.0	2.5	3.0	3.5	0.0	0.0	1.0
2	22	2.0	2.0	3.0	1.0	1.5	2.0	0.0	1.0	1.0
3	87	2.0	3.0	4.0	1.5	2.0	2.5	3.0	3.0	4.0
4	126	1.0	2.0	2.0	2.0	2.5	3.5	1.0	1.0	2.0
5	95	1.0	2.0	2.0	1.5	2.0	3.0	1.0	1.0	2.0
6	161	1.5	2.0	3.0	1.5	2.0	2.5	0.0	1.0	1.0
7	55	1.0	2.0	2.0	1.5	2.0	2.0	1.0	3.0	3.0
8	22	1.0	2.0	2.0	1.0	2.0	2.0	1.0	3.0	4.0
9	17	1.0	2.0	3.0	1.0	1.5	2.5	0.0	0.0	1.0
10	118	1.0	2.0	2.0	2.0	2.25	3.0	1.0	1.0	3.0
11	119	1.0	1.0	2.0	1.0	1.5	1.5	2.0	3.0	3.0
12	128	2.0	2.0	3.0	2.0	2.5	3.0	1.0	2.0	3.0
13	69	1.0	2.0	3.0	2.0	2.0	3.0	1.0	2.0	3.0
14	19	2.0	3.0	3.0	1.0	2.0	2.5	1.0	3.0	3.0
15	42	2.0	2.0	3.0	1.5	1.5	2.0	1.0	2.0	3.0
16	86	1.0	2.0	3.0	2.5	3.0	3.5	0.0	1.0	1.0

**Table 3 animals-13-03206-t003:** Descriptive statistics summarizing the cumulative egg shedding per farm (summary of EpG values of the entire sheep) and the number of animals contributing to the majority (>90% EpG percentile and >75% percentile EpG) of excreted eggs.

Farm ID	N Sheep	Cumulative Egg Shedding Ʃ Farm	>90% Percentile		>75% Percentile	
			N Sheep	Ʃ Eggs (%)	N Sheep	Ʃ Eggs (%)
1	29	5365	2	1670 (31.1)	7	3855 (71.9)
2	22	4110	2	1905 (46.4)	5	3040 (74.0)
3	87	98,670	8	42,540 (43.1)	21	71,545 (72.5)
4	126	27,135	12	15,980 (58.9)	29	22,730 (83.8)
5	95	158,245	9	64,685 (40.9)	23	106,435 (67.3)
6	161	212,195	16	109,455 (51.6)	39	159,990 (75.4)
7	55	23,855	5	8210 (34.4)	13	16,350 (68.5)
8	22	5590	2	2150 (38.5)	5	3495 (62.5)
9	17	24,990	1	4615 (18.5)	4	11,815 (47.3)
10	118	54,475	11	22,860 (42.0)	29	38,485 (70.6)
11	119	74,790	11	39,475 (52.8)	29	60,680 (81.1)
12	128	136,430	12	52,045 (38.1)	32	97,080 (71.2)
13	69	71,475	6	24,155 (33.8)	17	49,450 (69.2)
14	19	5980	1	2085 (34.9)	4	3870 (64.7)
15	42	14,810	3	5680 (38.4)	10	10,570 (71.4)
16	86	34,160	8	13,685 (40.1)	21	24,495 (71.7)

**Table 4 animals-13-03206-t004:** Results of the larval differentiation from pooled samples (n = 965 samples, n = 89 pools). Samples were pooled at the herd-level, taking into account the lactation number of the sheep. The table shows the percentage of *H. contortus* per farm classified after Knoll et al. (2021) [40].

Farm ID	N Samples	N Pools	% *H. contortus*
1	29	6	24.9
2	18	6	65.7
3	87	6	61.6
4	31	3	50.6
5	94	6	48.5
6	159	10	70.8
7	48	6	66.1
8	20	3	48.7
9	17	3	63.1
10	93	9	67.9
11	77	7	65.9
12	103	8	64.2
13	64	6	44.1
14	15	2	31.2
15	37	2	66.7
16	73	6	73.9

**Table 5 animals-13-03206-t005:** Results of the sedimentation and Baermann–Wetzel technique from pooled samples (n = 246 pools). Samples were pooled at the herd-level. The table shows the amount of positive pools with *D. dentriticum*, *F. hepatica* and Protostrongylidae per farm.

Farm ID	N Pools		N Pools Positive	
		*D. dentriticum*	*F. hepatica*	Protostrongylidae
1	6	0	0	3
2	6	0	0	1
3	17	9	0	17
4	26	9	0	3
5	19	5	0	0
6	34	0	0	11
7	12	0	0	5
8	6	5	1	5
9	4	0	0	4
10	23	6	0	4
11	23	0	9	0
12	25	0	0	25
13	14	0	0	0
14	4	3	0	3
15	9	3	0	8
16	18	0	0	2

**Table 6 animals-13-03206-t006:** The inter-rater agreements for the clinical parameters FAMACHA©, BCS, and dag score, calculated using the weighed Kappa coefficient. Rater 1 was a veterinarian, raters 2 and 3 final year veterinary students. Kappa values were interpreted as follows: ≥0.81 very good agreement; 0.61 to 0.80 good agreement; 0.41 to 0.60 moderate agreement; 0.21 to 0.40 fair agreement; and ≤0.2 poor agreement [43].

Clinical Parameter	Rater	N Sheep	Kappa	Lower 95%CI	Upper 95%CI
FAMACHA©	1vet–2stud	256	0.675	0.596	0.755
	1vet–3stud	232	0.667	0.579	0.756
	2stud–3stud	469	0.808	0.766	0.851
BCS	1vet–2stud	256	0.632	0.549	0.716
	1vet–3stud	233	0.669	0.600	0.738
	2stud–3stud	194	0.570	0.479	0.660
Dag score	1vet–2stud	209	0.718	0.637	0.799
	1vet–3stud	185	0.654	0.540	0.769
	2stud–3stud	148	0.616	0.491	0.741

**Table 7 animals-13-03206-t007:** Overview on the correlation (Spearman Rank correlation coefficient, r) between the excreted eggs per gram of faeces (EpG) and the clinical parameters (FAMACHA©, BCS, dag score) in 1195 dairy sheep from 16 farms in Austria. In bold *p* < 0.05.

Egg Excretion	N Farms	N Samples	Parameter	r (95%CI)	*p* Value
Total EpG	16	1195	FAMACHA©	0.196 (0.139; 0.251)	**<0.001**
			BCS	−0.156 (−0.212; −0.098)	**<0.001**
			Dag score	0.041 (−0.018; 0.099)	0.158
>50% *H. contortus*	11	961	FAMACHA©	0.233 (0.171; 0.294)	**<0.001**
			BCS	−0.149 (−0.212; −0.085)	**<0.001**
			Dag score	0.027 (−0.038; 0.092)	0.407
≤50% *H. contortus*	5	234	FAMACHA©	0.027 (−0.105; 0.158)	0.682
			BCS	−0.236 (−0.357; −0.107)	**<0.001**
			Dag score	0.133 (0.001; −0.261)	0.041

## Data Availability

Data are available within the article or its Appendix A.

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
