# Peer review of "Correlation of Faecal Egg Counts with Clinical Parameters and Agreement between Different Raters Assessing FAMACHA©, BCS and Dag Score in Austrian Dairy Sheep"

_animals, 2023, doi:10.3390/ani13203206_

Round 1

Reviewer 1 Report

The author report about the correlation of faecal egg counts with clinical parameters and agreement between different raters assessing FAMACHA©, BCS and dag score in Austrian dairy sheep.

The results were useful for farm management and the manuscript was well-written. I have only a minor comment;

In line 237, although the authors describe "The chi-square test was used on qualitative data." the data were not shown in result section. If you could not show any result, you should delete it.

Reviewer 2 Report

Comments on the article “Correlation of faecal egg count with clinical parameters and agreement between different raters assensing FAMACHA©, BCS and dag score in Austrian dairy sheep” by Sajovitz  al,  Animals - 2640402.

Sajovitz and colleagues in this study evaluated the relationship between certain clinical indicators in lactating dairy ewes and strongyle egg counts in faeces. In addition, the degree of agreement of three evaluators on the outcome of the clinical examination was evaluated. For this study, 1,195 dairy ewes from 16 farms were used. Individual subjects were evaluated for FAMACHA, BCS and dag scores and Mini-FLOTAC was used for parasitological diagnosis. Afterwards, faecal samples were processed for larval culture. Faecal egg count and BCS had a negative association, whereas the FAMACHA score and faecal egg count had a marginally favourable link. The Dag score did not reveal a clear link.  The conclusion reached is that the clinical score does not allow extrapolation of egg excretion from faeces.

The presented study is interesting, the experimental design is consistent and the manuscript is written in good English. I have no particular comments to make, but minor corrections are recommended to further improve its quality.

In particular, we recommend checking the italics throughout the text and bibliography.

In addition, many statements that are made are not always supported by bibliographical notes for reference. Therefore, it is recommended to add more bibliographical references.

Reviewer 3 Report

This study shows results about the correlation about the faecal nematode egg count (using Mini-FLOTAC method) and FAMACHA, body condition and dag scores in Austrian lactating dairy ewes. These data were obtained using individual samples. Likewise, faecal samples were pooled for larval culture according to the number of lactations of the farms. Additionally, the study included the participation of farmers that contributed supplying important information about a survey trough questionnaires about management practices in their farms, including diagnosis and control and the use of chemical anthelmintic drugs in their flocks.  

Their results showed a negative body condition score with the fecal egg count and a light positive correlation with FAMACHA scores. On the other hand, Dag score did not show a significant correlation. 

In my opinion, this is a valuable research that shows interesting results that are similar to other studies carried out under totally different environmental conditions ie., in tropics; where a small percentage of sheep of the whole flock concentrate the highest parasitic burden, elimination of eggs per g of faeces and a deplorable body conditions. I think this study was performed with a high standard scientific quality. One important part of this study is the information provided by farmers that reinforce the results of this study. Regarding the presence of blood-feeding nematode Haemonchus contortusit is interesting the fact that authors found a high prevalence of this highly pathogenic nematode reaching almost 60% per farm. 

I have some ideas that could help to authors to improve their manuscript. In my opinion some general information about the climate or meteorological conditions that prevail in the geographic area of study in Austria should be incorporated somewhere in the Introduction section of the manuscript. Also, it would be possible to include some information about the economic loses estimated in Austria?  Additionally, I think including some basic climatic or meteorological parameters ie., temperature, rainfall, humidity, etc. that occurred during the study it would be great to have a better conceptualization of the behavior of both the parasites and the ewes during the study.  

In my opinion this study offers a useful solid package of information that deserves to be published.

Mi verdict is: Minor Revision.
